# Mutations and Recombination at G4 DNA-Forming Sequences Exacerbated by CPT-Resistant Mutant Topoisomerase 1 Is Dependent on SUMOylation

**DOI:** 10.3390/ijms26189017

**Published:** 2025-09-16

**Authors:** Shivani Singh, Xinji Zhu, Nayun Kim

**Affiliations:** 1Department of Microbiology and Molecular Genetics, University of Texas Health Science Center at Houston, Houston, TX 77030, USA; 2Division of Pharmacology and Toxicology, College of Pharmacy, University of Texas at Austin, 1400 Barbara Jordan Blvd, Austin, TX 78723, USA

**Keywords:** topoisomerase 1, G4 DNA, genome instability

## Abstract

Topoisomerase 1 (Top1) removes transcription-related helical torsions and thus plays an important role in preventing genome instability instigated by the formation of non-canonical DNA secondary structures. The genetically tractable *Saccharomyces cerevisiae* model proved effective in defining the critical function of Top1 to prevent recombination and chromosomal rearrangement at G4-forming genomic loci and studying the human cancer-associated Top1 mutants through the expression of analogous yeast mutants. We previously showed that cleavage-defective Top1 mutants strongly elevate the rate of recombination at G4 DNA, which involves binding to G4 DNA and interaction with the protein nucleolin (Nsr1 in yeast). Here, we further explored the mechanism of genome instability induced by the yeast Top1Y740* mutant, analogous to the human Top1W765Stop mutant conferring resistance to CPT. We show that yTop1Y740* elevates duplications as well as recombination specifically at G4-forming sequences. Interestingly, SUMOylation of yTop1Y740*, which does not affect the G4 DNA-binding or Nsr1-interaction by this mutant, is necessary for such elevated G4-specific genome instability. Many tumors with mutations at the C-terminal residues of Top1, particularly W765, have significantly high G4-associated mutations, underscoring the importance of further investigation into how SUMOylation affects the function of these Top1 mutants at G4 DNA.

## 1. Introduction

Topoisomerase 1 (Top1) is a ubiquitously present enzyme that relieves helical stress by first binding to double-stranded (ds) DNA, then nicking one-strand of DNA with a catalytic tyrosine residue, and finally re-ligating the nicked DNA strand after controlled swiveling [1]. At genomic sites with repetitive sequences prone to formation of non-canonical secondary structures, Top1 importantly prevents the accumulation of torsional stress that can foster the DNA structure shifts, including the formation of G-quadruplex (G4) DNA from guanine-run-containing sequences. G4 DNA impedes dynamic cellular processes such as DNA replication and transcription and concomitantly leads to elevated genome instability [2,3,4,5]. Disruption of Top1 exacerbates the genome instability associated with G4 DNA formation. The absence of functional Top1 results in a particularly acute impact on genome instability at highly transcribed areas due to the accumulation of associated helical stresses, as shown by the siRNA-mediated knockdown in mouse B-cells or the gene deletion of Top1 in the *S. cerevisiae* model [2,6,7]. In addition to the suppression of G4-associated genome instability through removal of transcription-associated negative supercoils, the functional relevance of Top1 in the dynamics of G4 DNA was shown by the reported physical interaction between G4 DNA and human Top1 [8,9]. More recently, we showed that yeast Top1 also binds to G4 DNA formed from guanine-run-containing DNA oligos in vitro [10,11]. Top1 binding the G4 DNA present at the promoter of c-Myc gene as well as being trapped by co-transcriptionally-formed G4 DNA was also demonstrated in human embryonic kidney cell line Hek293T and human colon cancer cell line HCT116, respectively [12,13].

Human Top1 shares a high degree of homology with the yeast protein, especially at the enzymatically critical Core and C-terminal domains and can functionally complement the loss of yeast Top1 [14]. The catalytic cycle to topoisomerase 1, consisting of DNA binding, DNA cleavage via covalent DNA-protein complex formation, DNA strand swivel, and finally strand ligation, is also shared between human and yeast enzyme. Budding yeast, *Saccharomyces cerevisiae*, has been used as a model organism to study the mechanism of human Top1 function and identify and characterize drug-resistant variants of Top1 for decades [15,16,17]. In human cancers, Top1 can be targeted by anticarcinogenic drug camptothecin (CPT) and its analogs including irinotecan and topotecan (reviewed in [1]). CPT traps Top1 by intercalating into and trapping the Top1-cleavage complex (Top1cc) consisting of Top1 covalently attached to the 3′ end of a nicked DNA strand. There are, however, frequent incidences of drug-resistance in cancers treated with CPT or CPT-derivatives. The predominant mechanism of resistance appears to be the mutation of Top1 itself that disrupts the DNA binding or cleavage activity of the enzyme (reviewed in [10]).

We previously took advantage of the yeast model system to study the effect of two CPT-resistance-conferring Top1 mutants identified in human cancer cells [11], hTop1W763Stop mutation—resulting in the truncation of three C-terminal residues—which was identified in a non-small cell lung cancer patients treated with the CPT-derivative irinotecan [18]. This mutation results in a severe defect in the cleavage activity but does not affect the dsDNA binding activity of Top1. Mutations at T729 residue of human Top1 were first identified to confer resistance to irinotecan in CPT-treated human lung cancer line PC7-7 [19]. T729E mutant, in particular, was shown to be significantly defective in DNA binding. We constructed yeast strains expressing yTop1Y740* (analog to hTop1W763Stop) and yTop1S733E (analog to hTop1T729E) to study how DNA cleavage defect and DNA binding defect each affect the role of Top1 in suppressing G4-associated genomic instability. Our findings showed that, while these two mutants of Top1 similarly lack the conventional function of type 1B topoisomerase leading to CPT-resistance and to an inability to process DNA-embedded ribonucleotides, the impact on G4 DNA and its associated genome instability was clearly distinctive. The expression of cleavage-defective yTop1Y740* mutant but not of DNA-binding-defective yTop1S733E mutant results in sharply elevated recombination rates at a G4-forming sequence embedded in the yeast genome [11]. We showed that yTop1Y740* but not yTop1S733E binds to G4 DNA in vitro, postulating that complex formation of the functionally defective mutant Top1 with G4 DNA in the genome contributes to genome instability, instigating replication impediment. Our previous work further showed that the elevated genome instability of yTop1Y740* mutant requires interaction with yeast nucleolin Nsr1 and is partially suppressed by the SUMO-dependent protease Wss1, a homolog of human enzyme SPRTN that targets DNA–protein crosslinks.

Based on our previous findings, we here further characterized the effect of Top1Y740* mutant on the stability of the yeast genome. Our data show that the impact of Top1Y740* elevates recombination as well as mutations specifically at G4-DNA-forming regions of the genome, closely reflecting the frequent mutations at G4-forming sequences in cancer cells with mutations at C-terminal residues of human Top1. Additionally, eliminating the post-translational modification of yeast Top1Y740* mutant by SUMOylation abrogated the highly elevated genome instability at G4-forming sequences. We gained some insight into the mechanism by showing that elimination of SUMOylation does not affect the binding to G4 DNA or the interaction with another G4-binding protein Nsr1. Our work described here shows that the genetically tractable *S. cerevisiae* is a very useful model system to study the pathologically toxic human Top1 mutants arising in CPT-treated cancer cells and can lead to better understanding of the mechanism of G4-associated genome instability and potential secondary cancers linked to these drug-resistant mutant proteins.

## 2. Results

### 2.1. The Expression of Top1Y740* Mutant Affects Genome Instability Specifically at G4-Forming Sequences

Top1Y740* mutant was generated by introducing a C-terminal 3XFlag tag at the amino acid position 740 into the genomic copy of a haploid yeast strain. We first tested whether Top1Y740* can carry out topoisomerase function. Functional Top1 cleave at ribonucleotides remaining unexcised in DNA, thereby elevating mutations at genomic loci containing tandem repeats such as dinucleotide repeats [20,21]. When the mutation rate at the tandem-repeat (AGAGAGAG)-containing reporter in *rnh1∆ rnh201∆* strain background was measured, the highly elevated mutation rate in wt Top1-expressing yeast cells were all significantly reduced in *top1∆* background and in cells expressing Top1S733E or/and Top1Y740* mutants (Figure 1A). Next, we determined the rate of canavanine resistance (Can^R^) mutations in cells expressing wt or mutant Top1. Canavanine is a toxic analog of amino acid arginine taken up into the cell using the arginine transporter [22]. Mutations at the transporter-encoding *CAN1* gene resulting in canavanine resistance are one of the methods used to assess general mutagenic phenotypes of yeast strains. Since *CAN1* gene does not contain any G4-forming sequence, we used Can^R^ mutation rates to determine whether the disruption of Top1 affects mutation rates at non-G4 genomic loci. The Can^R^ rate in Top1y740*-expressing cells did not show any statistical difference when compared to that in wt Top1-expressing or *top1∆* cells (Figure 1B).

Recombination rates at the non-G4-forming sequence were determined using a genome-embedded reporter containing a portion of immunoglobulin switch region Mu (*Smu*) sequence integrated into the *pTET*-controlled *LYS2* gene in reverse orientation (*pTET-LYS2-SmuR*) [2]. In this orientation, the G4-forming guanine runs are in the non-coding (transcribed) strand rather than in the coding (non-transcribed) strand, where G4 formation is favored due to the transient single-strandedness. We previously reported on strand-specific elevation in recombination rate at this reporter in absence of Top1 [6,23]. As shown in Figure 1C, the expression of Top1Y740* resulted in no increase in the rate of recombination compared to wildtype cells, similar to *top1∆* and Top1S733E-expressing cells. We also measure the recombination rates at the similar reporter, with Smu sequence inserted in the orientation favoring G4 formation, with the guanine runs placed in the non-transcribed strand (*pTET-LYS2-SmuF*). At this G4-forming recombination reporter, *top1∆* and Top1S733E expression led to ~5- and 3.3-fold elevation in recombination rates, consistent with the previous report [11]. The expression of Top1Y740* mutant led to a close to 30-fold elevation in the recombination rate, which is significantly greater than the effect of *top1∆* or Top1S733E expression (Figure 1D). Overall, Top1Y7408 expression elevated recombination specifically at the G4-forming sequence while not affecting genome-wide mutagenesis or the recombination at the non-G4 sequence.

### 2.2. The Expression of Top1Y740* Mutant Elevates Duplication Events Proximal to G4-Forming Sequences

We next tested whether the expression of this G4-DNA-binding mutant Top1 affects deletion mutations occurring at a G4-forming sequence. We used yeast strains with a 130 bp fragment of human *DIP2C* gene, which was verified to form G4 conformation by circular dichroism analysis and primer extension assay, integrated into the yeast *LYS2* locus [24]. At this reporter (*pGAL-DIP2C*), *DIP2C* sequence is highly transcribed from the galactose-inducible *pGAL* promoter when cultured in media containing 2% galactose. Williams et al. previously showed that deletions or duplications of >10 nt occur at these sequences with significantly higher frequency compared to a control construct containing a similar length of GCA repeats. First, deletion of *TOP1* gene does not lead to a noticeable elevation in large duplications at *DIP2C* fragment or GCA repeats (Figure 1E,F). However, the expression of Top1Y740* in *top1∆* strain led to a modest increase in overall mutation rate for the DIP2C-containing reporter (Figure 1F) and specifically an increase in duplications > 10 nt from 20.6 × 10^−8^ to 30.4 × 10^−8^ (Table 1). For both *top1∆* and Top1Y740*-expressing cells, the duplications ranging from 19 to 49 nts in size occurred within the *DIP2C* insert and were flanked by microhomologies ranging from 5 to 11 nts (Figure 2 and Appendix A). The elevation in the duplication events in *DIP2C* fragment in Top1Y740*-expressing cells was relatively small compared to the elevation in recombination rates at Sµ fragment, potentially indicating that the mechanism leading to the duplication events and the recombination events might be distinct.

**Table 1 ijms-26-09017-t001:** Rate of large duplications at DIP2 fragment.

Genotype	Total Rate of Mutation (×10^−8^)	Total Mutants Sequenced	# of >10 nt Duplications	Rate of >10 nt Duplications (×10^−8^)
*TOP1*	25.0	45 *	38 *	21.1
*top1∆*	23.6	47	41	20.6
*top1Y740**	35.8	46	39	30.4

* From Williams et al. [24].

### 2.3. Recombination at G4 Is Significantly Reduced When the SUMOylation of Top1Y740* Is Abolished

We previously showed that Wss1 protease can partly suppress the elevation in G4-associated recombination affected by Top1Y740* expression [11]. Since Wss1 is targeted to SUMO-modified proteins, we accordingly confirmed that wildtype as well as mutant Top1 proteins are modified by SUMOylation. To determine the effect of the post-translational modification, we introduced mutations to wildtype and mutant Top1 to abrogate SUMOylation. K103, K117, and K153 of human Top1 were identified as sites of SUMO ligation and mutations of these lysines were shown to abolish SUMOylation [25]. We mutated the three lysines at putative SUMOylation sites in yTOP1 (K65, k91, and K92) [26] to arginine and confirmed the elimination of SUMOylation by SUMO pull-down (Appendix A and Figure 3A). For wildtype Top1 and Top1S733E, there was no change in the G4-associated recombination at the *pTET*-*LYS2-SmuF* reporter following KKK->RRR mutation. For Top1Y740*, the recombination rate was reduced by greater than 8-fold when SUMOylation was abolished (Figure 3B). We took a second independent approach to eliminate SUMOylation. Siz1 and its paralog Siz2 are E3 SUMO ligases responsible for the transfer of SUMO (Smt3) [27] and Top1-SUMO conjugation is eliminated in a *siz1∆ siz2∆* strain background [28]. As Figure 3C shows, the SUMOylation of wildtype, Top1S773E, or Top1Y740* were all significantly reduced in *siz1∆ siz2∆* backgrounds. Similar to KKK->RRR mutation, the deletion of *SIZ1* and *SIZ2* resulted in significant changes to the rate of recombination at the *pTET*-*LYS2-SmuF* reporter only in the Top1Y740*-expressing cells. In cells with functioning SUMO ligases, the G4-associated recombination rate in Top1Y740*-expressing cells was about 6-fold higher than that in *top1∆* background (Figure 3D). In *siz1∆ siz2∆* backgrounds, however, the rate of recombination in Top1Y740*-expressing cells was reduced to be similar to those in *top1∆* cells or Top1S733E-expressing cells. Overall, the drastic decreases in the rate of recombination at the *pTET*-*LYS2-SmuF* reporter observed upon disrupting SUMO ligases (*siz1∆ siz2∆* background) or mutating the SUMO ligation sites on Top1Y740* (KKK->RRR) indicate that SUMOylation of Top1Y740* is an important component of the mechanism of genomic instability at G4-containing sequences induced by the cleavage-defective Top1 mutant.

### 2.4. Binding to G4 DNA Is Not Affected When the SUMOylation Sites of Top1Y740* Are Mutated

We previously showed that Top1Y740* capable of binding to G4 DNA and not Top1S733E mutant with a defect in DNA binding elevates genome instability at a G4-containing reporter. Therefore, we postulated that the physical interaction of Top1 mutant with G4 DNA is necessary to instigate genome instability event. To determine whether SUMO modification of Top1 affects the binding to G4 DNA, we carried out an oligo pull-down assay using a G4-forming oligo with a G4Hunter score of 1.49 (GCTGGGCAGGTCAGGGCAGGGGCTCTCAGGGGGGCGC-G4 in Figure 4). This oligo represents a part of the G4-forming sequence embedded in the *pTET*-*LYS2-SmuF* reporter and, furthermore, verified to form a parallel G4 structure by circular dichroism analysis [29]. As a reference, cKit2 oligo CGGGCGGGCGCTAGGGAGGGT with four closely spaced guanine runs has the G4Hunter score of 1.57 [30]. As a negative control, an oligo where four of the guanines located within the guanine runs are replaced with adenine to disrupt G4 formation (MUT in Figure 4) was used. The G4Hunter score for the MUT oligo was 0.89. As previously demonstrated, both wildtype Top1 and Top1Y740* bind to the G4-forming oligo but not to the MUT oligo. Such differential binding was not affected by the KKK->RRR mutation or by the disruption of Siz1 and Siz2 (Figure 4B,C). The binding to the negative control oligo was also largely unaffected by either of the approaches used to eliminate SUMOylation of Top1. Overall, the abrogation of SUMOylation of Top1Y740*, which acutely decreases the G4-associated recombination rates, does not markedly affect G4 DNA binding by the mutant protein.

### 2.5. Interaction with Yeast Nucleolin Nsr1 Is Not Affected When the SUMOylation Sites of Top1Y740* Are Mutated

Yeast nucleolin Nsr1 is a protein with multiple functions that binds to G4 DNA and RNA through its C-terminal domain consisting of RNA recognition motifs (RRMs) and RGG repeats [31,32,33]. In top1∆ yeast cells, binding of Nsr1 to G4 DNA through the C-terminally located RGG repeats impedes replication and results in genome instability [34]. Additionally, Nsr1 physically interacts with Top1 through its N-terminal domain [35]. When the N-terminal domain of Nsr1 was truncated, the elevated instability at G4 in top1∆ background remained unaffected but the highly elevated genome instability in Top1Y740*-expressing cells was reduced to the level comparable to that in top1∆ cells [11,34]. This result indicated that the interaction between the two G4-DNA-binding proteins, Top1Y740* and Nsr1, is required for the induced recombination at G4-forming *pTET-LYS2-SmuF* reporter. Therefore, we tested whether the disruption of Nsr1-Top1Y740* interaction could be the underlying cause of greatly reduced G4-associated genome instability upon abrogation of SUMOylation of Top1Y740*. We used co-immunoprecipitation (co-IP) assays to determine whether the mutation of SUMOylation sites of Top1Y740* (KKK->RRR) affects its interaction with Nsr1 in vivo. Figure 5 and Appendix A show that KKK->RRR mutation does not affect the physical interaction of Nsr1 with either wildtype Top1 or Top1Y740*.

### 2.6. Mutations at the C-Terminal Residues of Human Top1 Correlate with High Incidence of Mutations at G4-Containing Loci Genome-Wide

We previously showed that mutations in Top1 correlate with high mutation frequencies, particularly at the potentially G4-forming sequences in the cancer genomes. Such pathological function of mutant Top1 could be the underlying cause of secondary cancer development documented in patients following treatment with CPT derivatives [36,37]. To determine whether the significant impact on the G4-associated genome instability is characteristic of Top1 mutants with changes in the C-terminal domain like human W736Stop and yeast Y740*, we surveyed the cancer mutation database obtained from the Catalogue of Somatic Mutations In Cancer (COSMIC) at https://cancer.sanger.ac.uk/cosmic through file CosmicMutantExport_v94.tsv (accessed on 12 September 2025). In a pool of 300 randomly chosen tumor samples, the median % mutations at G4 are less than 0.01 [11]. When the random sampling was shifted to include only those with more than 400 mutations in the genome in consideration of the generally high mutation rates in tumors containing Top1 mutations, the median is 0.65. But when we surveyed the 14 tumor samples with mutations in the human Top1 C-terminal domain residues 713 to 765 (Table 2), the median for the percentage of mutations at G4 is significantly higher at 1.151, with a minimum of 0 to a maximum of 4.330. These include two separate samples with mutation at W736 residue of Top1 (W736C), one malignant melanoma and the other urinary tract carcinoma, with 1.125% and 1.544% mutations at G4, respectively.

## 3. Discussion

The function of Top1 to suppress the formation of non-canonical DNA secondary structures including G4 DNA by removing negative helical stress has been shown through multiple reports [6,7,23]. Top1 tightly loops around duplex DNA and cleaves DNA strands by forming a phosphotyrosyl bond using tyrosine Y727 or Y723 in yeast or humans, respectively [38]. Top1 catalytic cycle is complete when the 5′-OH group attacks the Top1-DNA cleavage complex (Top1cc), releasing Top1 and restoring the original phosphodiester bond. Top1 is an essential cellular enzyme and historically important target of anti-cancer drugs. The cytotoxic effect of camptothecin (CPT) and synthetic analogs of CPT like topotecan and irinotecan functions by trapping Top1cc and preventing restoration of DNA backbone, but often results in drug resistance traceable to the mutations at Top1 protein itself [39]. Taking advantage of the high conservation between yeast and human Top1, we previously showed that yeast Top1Y740*, an analog of CPT-resistance-conferring human Top1W765Stop truncation mutant, leads to sharply exacerbated genome instability at G4 DNA [11]. The cleavage-defective Top1Y740* elevated the recombination occurring at G4-forming genomic sites by binding to G4 DNA and also potentially by forming a higher order complex with the yeast nucleolin Nsr1. Another interesting finding from our previous investigation was that the elevation of G4-associated recombination in Top1-Y740*-expressing cells was further aggravated upon disruption of Wss1, an isopeptidase recognizing SUMO-modified protein and involved in proteolytic removal of DNA–protein complexes (DPCs) including Top1cc [40,41].

We further tested whether Top1Y740* mutant can affect genome instability at non-G4-forming sequences. The rate of canavanine resistant mutations, a generally used measure of mutability [42], did not change with the null deletion of Top1 or with the expression of Top1Y740* (Figure 1). There was no change in the rates of mutation or recombination at highly transcribed mutation and recombination reporters *pGAL-GCA* and *pTET-LYS2-SmuR* with high GC contents [2,24]. In order to determine whether Top1Y740* affects genome instability at G4-forming sequences, we chose to employ two well-characterized reporter constructs previously used to study G4-induced recombination (*pTET-LYS2SmuF*) and deletions (*pGAL-DIP2C*) [2,24]. Both of these reporters contain inducible promoters to allow conditional induction of G4 DNA formation, the absence of tetrocycline for *pTET* and the presence of galactose for *pGAL*. Additionally, both of the reporters contain G4 motifs that were verified in vitro to form G4 structures [24,29]. Elevation in duplication mutations and recombination were noted at *pGAL-DIP2C* and *pTET-LYS2SmuF* reporters (Figure 1D,F), each with the insertions of G-run-containing sequences, indicating that the effect of Top1Y740* was specific to G4 DNA.

Wildtype and mutant Top1 are SUMOylated at residues K65, K91, and K92 (Figure 3) [26]. Interestingly, we found an important role of SUMOylation for the G4-specific genome instability induced by the G4-DNA-binding Top1 mutant Top1Y740*. When SUMOylation was abrogated either by the mutation of the three lysine residues or by the disruption of SUMO ligases Siz1 and Siz2, there was a substantial decrease in the rates of recombination in the Top1Y740*-expressing cells. The rate of recombination at the G4-forming reporter *pTET-LYS2-SmuF* was ~6-fold higher in the Top1Y740*-expressing cells than in *top1∆* cells. At the same reporter, there was no statistical difference between the rate of recombination in cells expressing Top1Y740* with K65R, K91R, K92R mutations and that in *top1∆* (Figure 3B). Similarly, the recombination rate at *pTET-LYS2-SmuF* in the Top1Y740*-expressing cells was also sharply reduced in the *siz1∆ siz2∆* background to the level comparable to that in *top1∆* cells. The deletion of SIZ1 and SIZ2 genes did not affect the rates of recombination in *top1∆* cells or in cells expressing the non-G4-binding Top1Y733E mutant. We checked whether binding to G4 DNA or interaction with Nsr1 was affected by the abrogation of SUMOylation. However, the mutation of K65, K91, and K92 to arginine did not markedly affect the binding to G4-forming oligo or interaction with Nsr1 as determined by the co-IP experiment (Figure 4 and Figure 5). We also confirmed that binding to G4 DNA did not considerably change in *siz1∆ siz2∆* backgrounds for wildtype or Top1Y740* mutants (Figure 4), although the SUMOylation of Top1Y740* could potentially contribute to a subtle change in the kinetics of G4 DNA binding, which was not uncovered due to the limitation of the oligo pull-down approach used here. Human Top1 is comparably SUMOylated at the N-terminally located K103, K117, and K153 residues [25]. Such post-translation modification by SUMOylation is functionally significant for the Top1 localization to nucleoli, as previously reported, and for the genome instability at G4-DNA-forming loci, as we report here.

Finally, the yeast model system enabled us to show that Top1 mutants identified in human cancer cells have detrimental and pathological functions beyond conferring resistance to CPT and CPT analogs. By expressing the yeast Top1Y740*, an analog of human Top1W765Stop mutant, we gained mechanistic insight into how the genome instability instigated by a G4-DNA-binding Top1 mutant is dependent on the post-translational modification by SUMOylation. The highly elevated mutations at G4-forming sequences in human tumors expressing Top1 alleles with C-terminally located mutations further underscore the significance of these insights regarding Top1Y740*.

## 4. Materials and Methods

### 4.1. Yeast Strain and Plasmid Construction

Yeast strains used in this study were derived from YPH45 strain (*MATa*, *ura3-52 ade2-101 trp1*∆*1*) [43]. The construction and design of the recombination reporter were previously described in detail [2]. Gene knock out for construction of deletion strains were carried out using the standard two-step allele replacement method. For the construction of K65R, K91R, K92R (RRR) mutant allele of *TOP1*, the *deletto perfetto* method was used [44]. In brief, first *URA3Kl* gene was inserted into the *TOP1* gene at nucleotide position +190 to 270 and then replaced by a PCR-generated DNA fragment containing mutations. Sequence of the mutations introduced is in Appendix A.

### 4.2. Determination of Recombination and Mutation Rates

For each fluctuation analysis, 12 individual 1-mL cultures were used [2,45]. The recombination or mutation rates as well as the 95% confidence intervals were calculated using the method of median. For the Can^R^ mutation rate determination, yeast cells were cultured in YEP media supplemented with 2% glycerol and 2% ethanol (YEP-GE) for 3 days and plated on either SCD-Leu for determination of total CFU or SCD − Arg + 60 mg/L canavanine for selection of Can^R^ mutants. For the recombination rate determination, yeast cells were cultured in YEP media supplemented with 2% glycerol and 2% ethanol (YEP-GE). After the 3-day incubation at 30 °C, cultures were diluted appropriately and then plated on agar plates containing either SCD-Leu for determination of total CFU or SCD-Lys for selection of Lys+ recombinants. For the mutation rate determination of DIP2C-insert-containing strains, cells were cultured in YEP media supplemented with 2% glycerol, 2% galactose, and 2% ethanol and plated on SCD-Leu for determination of total CFU and SC-Lys supplemented with 2% glycerol, 2% galactose, and 2% ethanol for the Lys+ mutants. For each mutation or recombination rate, at least 12 individual cultures were plated and 95% confidence intervals were calculated for statistical analysis [46].

### 4.3. Mutation Spectra

From individual cultures grown in 1-mL YEP media supplemented with 2% glycerol, 2% galactose, and 2% ethanol, a single mutant forming a colony on plates with SC-Lys supplemented with 2% glycerol, 2% galactose, and 2% ethanol was purified and genomic DNA was prepared using a lyticase lysis method. The relevant region of *LYS2* containing DIP2C insert was amplified using primers LYSWINDF (5′-GCCTCATGATAGTTTTTCTAACAAATACG) and LYSWINDR (5′-CCCATCACACATACCATCAAATCCAC) and sequenced by Eurofins (Luxembourg)/Operon (Huntsville, AL, USA).

### 4.4. Western Blotting

Yeast whole cell lysates were prepared for Western blotting using the NaOH method as previously described [47]. Whole-cell lysate samples were resolved on 4–20% SDS-PAGE gels (Bio-Rad; Cat# 456-1093, Bio Rad Laboratories, Hercules, CA, USA) and transferred to PVDF membrane and then probed with either an HRP-conjugated α-FLAG antibody (Sigma; Cat# A8592, Sigma-Aldrich, St. Louis, MO, USA) or HRP-conjugated α-HA antibody (Sigma; Cat# H6533) as indicated in the respective figures. Blots were visualized using a Bio-Rad ChemiDoc™ MP imaging system and Top1 and Nsr1 protein levels were quantified using ImageJ v1.51. Where indicated, at least 3 independent experiments were performed to calculate statistical differences using a student’s *t*-test.

### 4.5. The Oligo Pull-Down Assay

In vitro oligo pull-down assays were performed as previously described in [29] with some modifications. For yeast whole-cell lysate preparation, yeast cell cultures were grown in YEP media with 2% glucose (YEPD) and collected at an OD_600_ of 0.8–1.0. For each sample, 400 pmol of biotinylated oligos with 3′ biotin-TEG modification purchased from Sigma were folded in 100 µL buffer containing 100 mM KCl and added to 100 µL of Streptavidin-Coupled M-280 Dynabeads (Invitrogen; Cat# 11205D, Invitrogen Corporation, Carslbad, CA, USA). The Western blotting analysis of pulled down and input samples (cell lysate) were carried out as described above.

### 4.6. Top1-SUMO Pull-Down

Pull-down of SUMO-modified Top1 was carried out as described previously [11]. pGPD2-His-SMT3 was transformed into yeast strains harboring the indicated *TOP1* alleles. The pull-down of SUMO-conjugated proteins from whole-cell lysates were carried out using HisPur Ni-NTA Resin (Thermo Scientific; Cat# 88221, Thermo Fisher Scientific, Waltham, MA, USA). Wildtype or mutant Top1 in the input and pull-down samples were analyzed by Western blotting using an α-FLAG-HRP antibody (Sigma; Cat# A8592) and visualized using a Bio-Rad ChemiDoc MP imaging system.

### 4.7. Co-IP

Co-IP was carried out as described previously [11]. Briefly, yeast cells with Ha-tagged Nsr1 and Flag-tagged Top1 or Top1Y740* were cultured in YEPD. Co-IP samples were resolved on a 4–20% SDS PAGE gel (Bio-Rad; Cat# 4561093) and subjected to Western blotting. Three independent experiments were carried out followed by quantitative and statistical analyses.

## Figures and Tables

**Figure 1 ijms-26-09017-f001:**
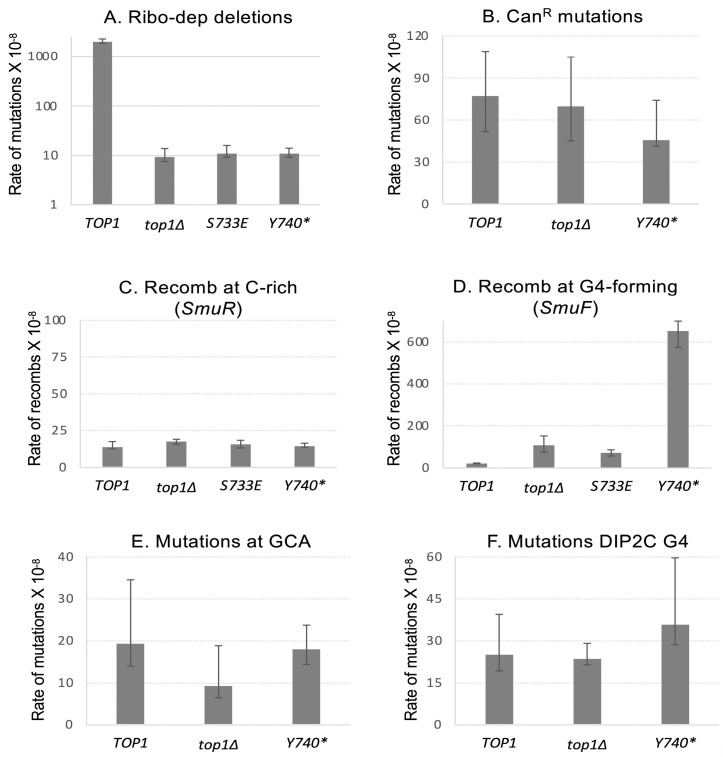
Rates of mutation and recombination in Top1Y740*-expressing cells. (**A**). The rates of deletion mutations at pTET-LYS2-AG4 reporter in rnh1∆ rnh201∆ backgrounds with the indicated Top1 alleles. (**B**). The rates of canavanine resistance mutations in the yeast strains with indicated Top1 alleles. (**C**). The rates of recombination at the pTET-LYS2-SmuR reporter with the C-run-containing strand in the non-transcribed strand orientation. (**D**). The rates of recombination at the pTET-LYS2-SmuF reporter with the G4-forming sequence in the non-transcribed strand orientation. (**E**). The rates of mutations at the GCA-repeat-containing reporter. (**F**). The rates of mutations at the pGAL-DIP2C reporter. (**A**–**F**): Error bars indicate 95% confidence intervals).

**Figure 2 ijms-26-09017-f002:**
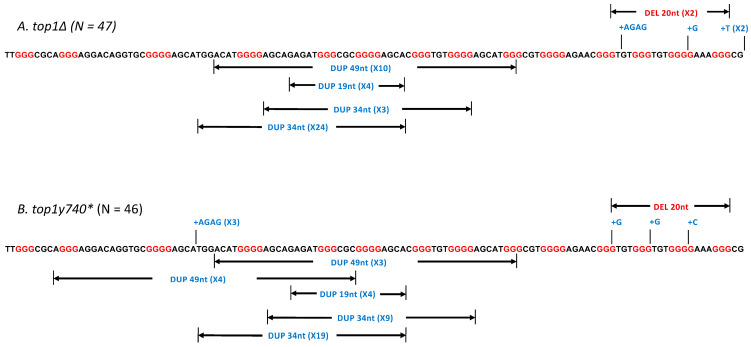
Mutation spectra at the *pGAL-DIP2C* reporter in *top1∆* and Top1Y740*-expressing cells. The sequence of *DIP2C* inserted into the *pGAL-DIP2C* reporter is shown in bold with the guanine runs indicated in red. The spectrum of mutants sequenced are shown. The locations of duplicated and deleted sequences are marked by the arrows and insertion mutations are indicated with +. When the same mutant sequences are identified multiple times, they are indicated by the numbers in (). 47 (**A**) and 46 (**B**) mutants are sequenced from *top1∆* and Top1Y740*-expressing strains, respectively.

**Figure 3 ijms-26-09017-f003:**
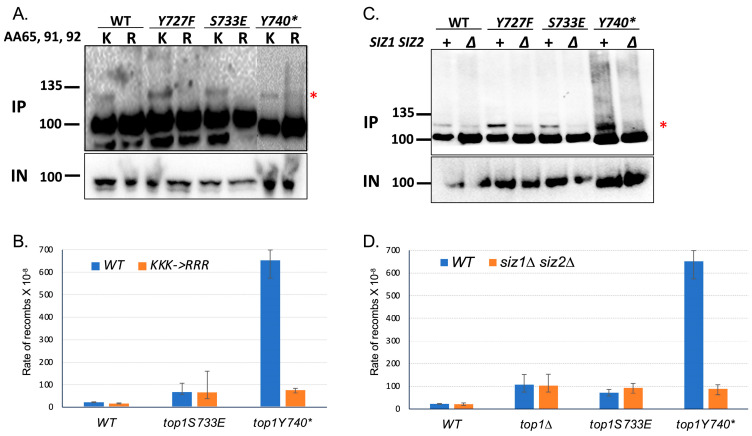
The effect of eliminating SUMOylation of Top1Y740* on the rates of recombination at G4. (**A**). SUMO pull-down assay. At Top1 residues 65, 91, and 92, wildtype or mutated amino acid sequence are K or R, respectively. SUMO-modified, upshifted bands are indicated with a red asterisk. Cell extracts used in experiments are from strains in wss1∆ backgrounds and expressing Top1 tagged with 3XFLAG at C-term and Smt3 tagged with 7XHis at N-term. Pull-down of SUMO-modified proteins were carried out using Ni+ beads. Proteins detected by Western blotting with α-FLAG-HRP antibody. Top panel—pull-down samples (IP); bottom panel—inputs (IN). (**B**). Recombination rates at the G4-containing pTET-LYS2-SmuF reporter in cells expressing wildtype, Top1S733E, or Top1Y740*. Blue bars are rates from wildtype or mutant Top1 with K65, K91, and K92. Orange bars are rates from wildtype or mutant Top1 with K65R, K91R, and K92R mutations. Error bars indicate 95% confidence intervals. (**C**). SUMO pull-down assay. Pull-down experiments were carried out with wildtype or indicated Top1 mutant-expressing cells in either SIZ1 SIZ2 or siz1∆ siz2∆ backgrounds as indicated. Other details are same as (**A**). (**D**). Recombination rates at the G4-containing pTET-LYS2-SmuF reporter in cells expressing indicated Top1 alleles in either SIZ1 SIZ2 or siz1∆ siz2∆ backgrounds.

**Figure 4 ijms-26-09017-f004:**
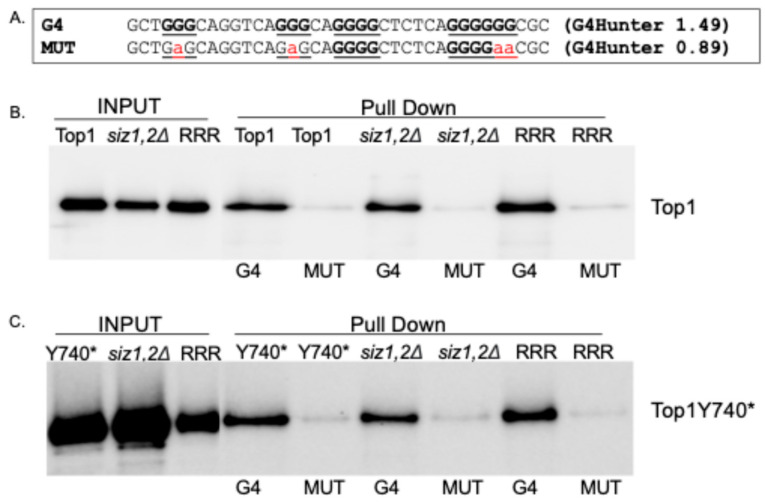
The effect of eliminating SUMOylation of Top1Y740* on G4 DNA binding. (**A**). The sequences and G4 Hunter scores for oligos used for the oligo pull-down assay. (**B**,**C**). Yeast whole-cell lysates from the indicated strains expressing WT or mutant Top1 proteins tagged with 3XFLAG were incubated with G4 or MUT biotinylated oligos and pulled down with streptavidin magnetic beads. Western blots of input and pull-down fractions were probed with an α-FLAG-HRP antibody (Sigma). Input and pull-down samples from wildtype Top1 in a SIZ1 SIZ2 background, wildtype Top1 in a siz1∆ siz2∆ background, and Top1K65R, K91R, K92R mutants in a SIZ1 SIZ2 background are shown in (**B**). Input and pull-down samples from Top1Y740* in a SIZ1 SIZ2 background, Top1Y740* in a siz1∆ siz2∆ background, and Top1Y740*, K65R, K91R, K92R mutants in a SIZ1 SIZ2 background are shown in (**C**).

**Figure 5 ijms-26-09017-f005:**
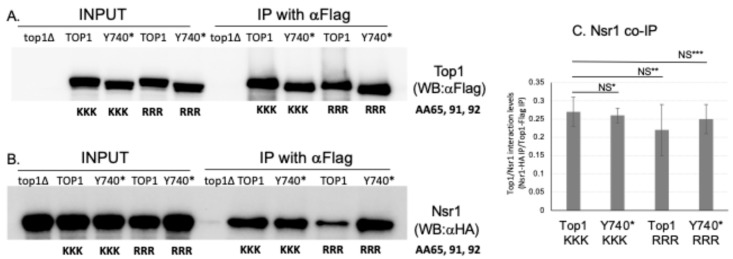
The effect of eliminating SUMOylation of Top1Y740* on Nsr1 interaction. Co-immunoprecipitation (co-IP) experiments conducted with whole-cell extracts from *vtc4∆* yeast strains expressing Top1-3XFLAG and Nsr1-6XHA. αFLAG antibody-coated agarose beads were used in pull-down. (**A**,**B**) are from one of three independent experiments carried out. The blots from the other two experiments are in Appendix A. Blots were probed with either αFLAG (**A**) or αHA (**B**) antibodies. (**C**). ImageJ v1.51 was used for the quantification of bands. Error bars represent standard deviations calculated from three independent experiments carried out. *p*-values were calculated using Student *t*-test. NS *—0.90, NS **—0.82, and NS ***—0.39.

**Table 2 ijms-26-09017-t002:** Tumors with mutations at C-terminal domain of Top1.

Sample Label	Mutation at Top1	Tissue	# of Mut Total	# of Mut at G4	% Mut at G4
T90	T747P	Prostate	49	1	2.041
APGI-DA-3191	T747N	Small intestine	85	1	1.176
TCGA-AD-6901-01	R749Q	Large intestine	319	0	0
TCGA-ER-A19E-06	W736C	Skin	711	8	1.125
TCGA-D3-A2JP-06	W754L	Skin	4857	6	0.123
TCGA-BS-A0UV-01	A715V	Endometrium	9959	16	0.161
SNU-C2B	I714T	Large intestine	3413	53	1.553
TCGA-D8-A27H-01	Q713H	Breast	184	0	0
sysucc-783T	A759T	Large intestine	1455	63	4.330
1117	R727S	Urinary tract	182	2	1.099
CHG-13-27889T	A753S	Liver	3948	109	2.761
PD36792a	I743T	Skin	1978	20	1.011
TCGA-G2-A2EJ-01	W736C	Urinary tract	907	14	1.544
163	W732C	Stomach	244	5	2.049

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
