# Peer review of "Mutations and Recombination at G4 DNA-Forming Sequences Exacerbated by CPT-Resistant Mutant Topoisomerase 1 Is Dependent on SUMOylation"

_ijms, 2025, doi:10.3390/ijms26189017_

Round 1

Reviewer 1 Report

Comments and Suggestions for Authors This manuscript (ijms-3856068) presents an interesting study of the interplay between
the camptothecin-resistant mutant Top1, G-quadruplex structures and protein SUMOylation
that determine the level of genetic instability in malignant cells. The authors used a variety of
modern powerful experimental approaches to address the challenges posed. Although the
manuscript is well written and the results support the
authors’ conclusions, some issues need to
be addressed before publication.

Major comments:

1. This work analyzes too many factors, making it difficult to differentiate them. The main findings
in each section should be presented more clearly.
2. The choice of G4-forming DNA sequences was not justified. It remains unclear why different
G4 motifs were used in
different sections; in section 2.1 the immunoglobulin switch region Mu,
in section 2.2 –
a fragment of the DIP2C gene, and in section 2.4 - G4-forming oligo with
G4Hunter score of 1.49
d(GCTGGGCAGGTCAGGGCAGGGGCTCTCAGGGGGGCGC)?
3. G4-folding potential of the G4-forming DNA sequences used in this work has been characterized
only in
directly. Direct evidence of the G4 formation by the corresponding single-stranded DNA
sequences can be easily made
in vitro, for example, by CD spectroscopy. The same method can
be
also used to determine the topology of G-quadruplexes - parallel or antiparallel. This is of crucial importance, since only parallel G4s, due to their geometric parameters, can be integrated into the DNA double helix [Kreig, A., Calvert, J., Sanoica, J., Cullum, E., Tipanna, R., and Myong, S. (2015) G-quadruplex formation in double strand DNA probed by NMM and CV fluorescence, Nucleic Acids Res., 43, 7961-7970].It should be noted that in Ref [24] it was shown by CD spectroscopy that the DIP2C gene
fragment folds into
G4 with a mixed parallel-antiparallel topology. A similar conclusion can be
made
by analyzing the primary structures of the DIP2C gene fragment and the G4-forming oligo
with G4Hunter score of 1.49
: too long spacers between G-tracts (potential loops) contribute to
the formation of
G4s with antiparallel or mixed topologies [Bugaut, A., and Balasubramanian, S.
(2008) A sequence-
independent study of the influence of short loop lengths on the stability and
topology of intramolecular DNA G
-quadruplexes, Biochemistry, 47, 689-697]. .

Minor comments:

1. Figure 2 is not cited in the main text.

2. Any fragments of the genome (e.g. DIP2C, cKit2, etc) should be written in italics. If they refer to the human genome, they should be written in capital letters.

3 Figure 4A. In the top line (G4), the sequence GGGGGG should be bolded.

Author Response

We thank the reviewer for the careful reading of the manuscript and the overall positive assessment. Our responses to the comments are in Bold below. 

Reviewer 1

This manuscript (ijms-3856068) presents an interesting study of the interplay between
the camptothecin-resistant mutant Top1, G-quadruplex structures and protein SUMOylation that determine the level of genetic instability in malignant cells. The authors used a variety of modern powerful experimental approaches to address the challenges posed. Although the manuscript is well written and the results support the authors’ conclusions, some issues need to be addressed before publication.

Major comments:

  1. This work analyzes too many factors, making it difficult to differentiate them. The main findings in each section should be presented more clearly.

Thank you for this comment.  We added a concluding sentence to summarize and clarify the findings to the parts of the RESULT section that was missing such sentence.

  1. The choice of G4-forming DNA sequences was not justified. It remains unclear why different G4 motifs were used in different sections; in section 2.1 – the immunoglobulin switch region Mu, in section 2.2 – a fragment of the DIP2Cgene, and in section 2.4 - G4-forming oligo with G4Hunter score of 1.49 d(GCTGGGCAGGTCAGGGCAGGGGCTCTCAGGGGGGCGC)?

We now added more explanation in the text for choosing the recombination reporter pTET-LYS2-Smu and the mutation reporter pGAL-LYS2-DIP2C. Briefly, the main justification is that they are both previously reported to be utilized to assess G4-associated genome instability and that the G4-formation of these sequences were previously confirmed using CD.  The G4 oligo used in section 2.4 is a sequence taken from the IG switch region Mu sequence integrated into the pTET-LYS2-Smu reporter and previously confirmed to form G4 structure.  These descriptions are now added to the text.

  1. G4-folding potential of the G4-forming DNA sequences used in this work has been characterized only indirectly. Direct evidence of the G4 formation by the corresponding single-stranded DNA sequences can be easily made in vitro, for example, by CD spectroscopy. The same method can be also used to determine the topology of G-quadruplexes - parallel or antiparallel. This is of crucial importance, since only parallel G4s, due to their geometric parameters, can be integrated into the DNA double helix [Kreig, A., Calvert, J., Sanoica, J., Cullum, E., Tipanna, R., and Myong, S. (2015) G-quadruplex formation in double strand DNA probed by NMM and CV fluorescence, Nucleic Acids Res., 43, 7961-7970].It should be noted that in Ref [24] it was shown by CD spectroscopy that the DIP2Cgenef ragment folds into G4 with a mixed parallel-antiparallel topology. A similar conclusion can be made by analyzing the primary structures of the DIP2C gene fragment and the G4-forming oligo with G4Hunter score of 1.49: too long spacers between G-tracts (potential loops) contribute to the formation of G4s with antiparallel or mixed topologies [Bugaut, A., and Balasubramanian, S. (2008) A sequence-independent study of the influence of short loop lengths on the stability andtopology of intramolecular DNA G-quadruplexes, Biochemistry47, 689-697]. 

Please see the response for 2. Furthermore, whether the effect of Top1 mutant is limited to the parallel G4 or not is an interesting question that should be further studied.  However, we find this to be beyond the scope of current manuscript.

Minor comments:

  1. Figure 2 is not cited in the main text. Corrected.
  2. Any fragments of the genome (e.g. DIP2C,cKit2, etc) should be written in italics. If they refer to the human genome, they should be written in capital letters. Corrected. 

3 Figure 4A. In the top line (G4), the sequence GGGGGG should be bolded. Corrected. 

Reviewer 2 Report

Comments and Suggestions for Authors

Comments

This interesting study shows that the yeast Top1Y740* mutant, analogous to human Top1W765Stop, increases duplications and recombination at G4 sequences, with SUMOylation contributing to this effect. Overall, the study is well designed and mechanistically relevant. I provide the following comments for the authors’ consideration.

Major comments:

  1. In the co-IPexperiments (Figure. 3A and 3C), I noticed that the input levels are not well balanced across samples. This makes it difficult to judge whether the differences observed in SUMO-Top1 bands reflect true biology or are partly due to loading variation. Including a standard loading control (e.g., Pgk1, Act1) in the input panels would strengthen the interpretation. In addition, the apparent molecular weight of SUMO-Top1 is indicated at different positions in Fig. 3A (~135 kDa) and Fig. 3C (<135 kDa), which might cause confusion to the reader.
  2. The input bands in the pull-down assay (Figure. 4C) appear unbalanced, so comparison between pull-down efficiency is problematic. It would be helpful if the authors could clarify the loading strategy or provide more balanced input controls.

Additionally, in the section (The oligo pull-down assay) of Method section, the G4 oligos were characterized as having been folded in buffer with either 100 mM KCl or LiCl, but the figure legends do not indicate which condition was used, and no LiCl control is shown. Since KCl stabilizes G4, while LiCl prevents G4 folding to serve as the standard negative control, it would strengthen the conclusions if the legends specify the ionic conditions used and, if available, LiCl control results are provided (or a reason is given if not provided).

  1. Comment on Figure 5 (Top1–Nsr1 co-IP assay):

Overall, the co-IP experiment is convincing and supports the interaction of Top1 and Nsr1. However, a few clarifications would strengthen the conclusion:

  • It is not clear whether panels A–C were generated from the same lysatesand co-IP experiment or from independent assays. If they are from the same source, please indicate this in the legend for clarity.
  • In panel A, the Top1–RRR IP signal looks reduced apparently compared to Top1–KKK, whereas in panel C the quantification suggests no significant difference. This apparent discrepancy could be due to experimental variability.Repeated experiment need to be done to clarify whether SUMOylation affects the interaction between Top1 and Nsr1.
  • In panel C, the third bar is mislabeled as “Top1 KKK” but should be “Top1 RRR.” Correcting this would avoid confusion.
  • The Methodsmentions the use of a t-test, but p-values are not shown. Even if the differences are not significant, reporting the values (or explicitly noting “ns”) would improve transparency.
  1. In the Discussion (page 10, paragraph 300~306), the authors state that Top1Y740* elevated G4-associated recombination “by forming a higher-order complex with Nsr1”. However, the presented data (Figure 5) only show that Top1 and Nsr1 can interact physically, without demonstrating that this interaction is mechanistically linked to the observed increase in recombination. I suggest that the authors rephrase this part more cautiously (e.g., “potentially through interaction with Nsr1”) and clearly distinguish between their own findings and previously published results (e.g., Wss1-related data).
  2. In both the Introductionand Discussion, the authors highlight the potential role of the SUMO-dependent protease Wss1 in regulating Top1Y740*-associated genome instability. However, no experimental data on Wss1 (e.g., in wss1Δ strains) are presented in this manuscript. The current phrasing (e.g., “we further showed…” in the Introduction) could give readers the impression that Wss1 was directly tested here, which is not the case. Similarly, the Introduction statement “We further showed that the elevated genome instability by yTop1Y740* mutant requires interaction with yeast nucleolin Nsr1” is not supported by data in this study. I suggest clarifying that these points are based on previous work, or alternatively including supporting experiments
  3. In Fig. 4, the authors show that SUMOylation does not affect Top1Y740* binding to G4 DNA in vitro, while in vivo SUMOylation deficiency suppresses Top1Y740*-induced recombination. This apparent discrepancy may be due to SUMOylation-dependent nucleolar localization of Top1(as mentioned in the Discussion Line 335~337), where rDNA repeats are G4-rich. Without SUMOylation, Top1Y740* may not accumulate at these hotspots, thereby reducing recombination. It would be useful for the authors to briefly discuss this mechanistic distinction.

Minor comments:

  1. In Figures 1A–F, Figure 3B and 3D, since the authors used fluctuation assays with ≥12 parallel cultures and reported 95% CI, it might help readers if the figure legends explicitly state that the error bars represent 95% CI.
  2. On Page 4 Line 110 “When the mutation rate at the tandem-repeat containing mutation...”, it would be more readable if add “G4 forming sequence” after “tandem-repeat”.
  3. On Page 4 Line 140~144, please include the figure labeling (Figure 1D) for the results description.
  4. On Page 4 Line 163, “microhomologies ranging from 5 to 11 nts (Figure 3 and Table S1)” should be Figure 2.
  5. On Page 7Line 230~232, “Such differential binding was not affected by theKKK->RRR mutation or by the disruption of Siz1 and Siz2. The binding to the negative control oligo was also unaffected by either of the approaches used to eliminate SUMOylation of Top1.” should be labeled with Figure 4B and 4C.
  6. The rationale for testing Top1–Nsr1 interaction is not clearly introduced in the manuscript. While the experiments themselves are well conducted, it would help readers if the Introduction (or Results section leading text) briefly summarized the background of Nsr1 to support the readers to understand wether or how Nsr1 affect Top1Y740*-mediated mutations at G4 forming sequences. Otherwise, the sudden transition to the co-IP assay in Figure 5 may seem disconnected from the earlier results.
  7. On Page 9 Line 276, the phrase “less than <0.01” is redundant. Please revise to either “less than 0.01” or “<0.01” for clarity.

Author Response

We thank the reviewer for the careful reading of the manuscript and the overall positive assessment. Our responses to the comments are in bold below. 

Reviewer 2

This interesting study shows that the yeast Top1Y740* mutant, analogous to human Top1W765Stop, increases duplications and recombination at G4 sequences, with SUMOylation contributing to this effect. Overall, the study is well designed and mechanistically relevant. I provide the following comments for the authors’ consideration.

Major comments:

  1. In the co-IPexperiments (Figure. 3A and 3C), I noticed that the input levels are not well balanced across samples. This makes it difficult to judge whether the differences observed in SUMO-Top1 bands reflect true biology or are partly due to loading variation. Including a standard loading control (e.g., Pgk1, Act1) in the input panels would strengthen the interpretation.

Thank you for pointing this out.  We agree that the oligo pull down approach we report on in Figure 3 is not adequate to quantitatively assess the binding.  Thus, we tried to make sure not to overreach in our conclusion and reworded our statement to qualitative statement that there is no substantial change in SUMOylation. 

In addition, the apparent molecular weight of SUMO-Top1 is indicated at different positions in Fig. 3A (~135 kDa) and Fig. 3C (<135 kDa), which might cause confusion to the reader. Corrected.

  1. The input bands in the pull-down assay (Figure. 4C) appear unbalanced, so comparison between pull-down efficiency is problematic. It would be helpful if the authors could clarify the loading strategy or provide more balanced input controls.

We believe that results shown in Figure 4 shows that there is no substantial change in G4 binding.  We now added qualifying statement in the discussion that “the SUMOylation of Top1Y740* could potentially contribute to a subtle change in the kinetics of G4 DNA binding, which was not uncovered due to the limitation of the oligo pulldown approach used here.”

Additionally, in the section (The oligo pull-down assay) of Method section, the G4 oligos were characterized as having been folded in buffer with either 100 mM KCl or LiCl, but the figure legends do not indicate which condition was used, and no LiCl control is shown. Since KCl stabilizes G4, while LiCl prevents G4 folding to serve as the standard negative control, it would strengthen the conclusions if the legends specify the ionic conditions used and, if available, LiCl control results are provided (or a reason is given if not provided).  Thank you for pointing out this error.  We use two different types of negative control to show that the binding is G4 specific.  One is to compare the same oligos folded in K+ vs. Li+.  And the other is to compare an oligo with G-runs with a similar oligo with selected mutations to disrupt a G-run.  In this manuscript Figure 4, we took the second approach to compare binding to G4 oligo with binding to MUT oligo.  We have corrected Method section accordingly. 

  1. Comment on Figure 5 (Top1–Nsr1 co-IP assay):

Overall, the co-IP experiment is convincing and supports the interaction of Top1 and Nsr1. However, a few clarifications would strengthen the conclusion:

  • It is not clear whether panels A–C were generated from the same lysate sand co-IP experiment or from independent assays. If they are from the same source, please indicate this in the legend for clarity.

Figure 5A and 5B represents an experiment from the same set of lysates. Two additional set of experiments were carried out and the quantification of all three set of experiments are presented as a graph in Figure 5C.  We now added a supplemental Figure S1 that contains the two additional set of co-IP experiments. 

  • In panel A, the Top1–RRR IP signal looks reduced apparently compared to Top1–KKK, whereas in panel C the quantification suggests no significant difference. This apparent discrepancy could be due to experimental variability. Repeated experiment need to be done to clarify whether SUMOylation affects the interaction between Top1 and Nsr1.

We carried out three independent co-IP experiments and observed some fluctuation due to experimental variability.  Figure 5C presents the average of the three experiments with statistical analyses to show that there are no significant difference among the four strain backgrounds tested. 

  • In panel C, the third bar is mislabeled as “Top1 KKK” but should be “Top1 RRR.” Correcting this would avoid confusion. Corrected.
  • The Methodsmentions the use of a t-test, but p-values are not shown. Even if the differences are not significant, reporting the values (or explicitly noting “ns”) would improve transparency. We now noted “ns” in the graph and added the p-values in the legend.

  1. In the Discussion (page 10, paragraph 300~306), the authors state that Top1Y740* elevated G4-associated recombination “by forming a higher-order complex with Nsr1”. However, the presented data (Figure 5) only show that Top1 and Nsr1 can interact physically, without demonstrating that this interaction is mechanistically linked to the observed increase in recombination. I suggest that the authors rephrase this part more cautiously (e.g., “potentially through interaction with Nsr1”) and clearly distinguish between their own findings and previously published results (e.g., Wss1-related data).  We made the suggested wording changes to make it clear that it is our previous finding that showed Top1 mutant elevate G4-associated recombination potentially through interaction with Nsr1. Also, we now added more detailed description of our previous finding that led us to the conclusion that Top1mutant-Nsr1 interaction is important for the elevation of G4-associated recombination. 

  1. In both the Introductionand Discussion, the authors highlight the potential role of the SUMO-dependent protease Wss1 in regulating Top1Y740*-associated genome instability. However, no experimental data on Wss1 (e.g., in wss1Δ strains) are presented in this manuscript. The current phrasing (e.g., “we further showed…” in the Introduction) could give readers the impression that Wss1 was directly tested here, which is not the case. Similarly, the Introduction statement “We further showed that the elevated genome instability by yTop1Y740* mutant requires interaction with yeast nucleolin Nsr1” is not supported by data in this study. I suggest clarifying that these points are based on previous work, or alternatively including supporting experiments. We changed wordings in the INTRODUCTION to better separate our previous findings from our current findings, which includes the experimental data on Wss1.

  1. In Fig. 4, the authors show that SUMOylation does not affect Top1Y740* binding to G4 DNA in vitro, while in vivo SUMOylation deficiency suppresses Top1Y740*-induced recombination. This apparent discrepancy may be due to SUMOylation-dependent nucleolar localization of Top1(as mentioned in the Discussion Line 335~337), where rDNA repeats are G4-rich. Without SUMOylation, Top1Y740* may not accumulate at these hotspots, thereby reducing recombination. It would be useful for the authors to briefly discuss this mechanistic distinction. As we are assessing the recombination rate at a defined reporter rather than assessing genome wide genome instability, the nucleolar localization of Top1y740* is not likely to be relevant to our findings. However, although beyond the scope of this manuscript, we will take the reviewer’s suggestion to test the sub-nuclear localization of Top1y740* and Top1Y740*KKK->RRR. 

Minor comments:

  1. In Figures 1A–F, Figure 3B and 3D, since the authors used fluctuation assays with ≥12 parallel cultures and reported 95% CI, it might help readers if the figure legends explicitly state that the error bars represent 95% CI. Corrected.
  2. On Page 4 Line 110 “When the mutation rate at the tandem-repeat containing mutation...”, it would be more readable if add “G4 forming sequence” after “tandem-repeat”. This tandem-repeat is not G4-forming sequence.  We now added more description of the reporter for clarification.
  3. On Page 4 Line 140~144, please include the figure labeling (Figure 1D) for the results description.Corrected.
  4. On Page 4 Line 163, “microhomologies ranging from 5 to 11 nts (Figure 3 and Table S1)” should be Figure 2. Corrected.
  5. On Page 7Line 230~232, “Such differential binding was not affected by theKKK->RRR mutation or by the disruption of Siz1 and Siz2. The binding to the negative control oligo was also unaffected by either of the approaches used to eliminate SUMOylation of Top1.” should be labeled with Figure 4B and 4C.Corrected.
  6. The rationale for testing Top1–Nsr1 interaction is not clearly introduced in the manuscript. While the experiments themselves are well conducted, it would help readers if the Introduction (or Results section leading text) briefly summarized the background of Nsr1 to support the readers to understand wether or how Nsr1 affect Top1Y740*-mediated mutations at G4 forming sequences. Otherwise, the sudden transition to the co-IP assay in Figure 5 may seem disconnected from the earlier results. We now describe more fully the rationale for conducting this experiment.  This is added to the beginning of RESULT section 2.5.
  7. On Page 9 Line 276, the phrase “less than <0.01” is redundant. Please revise to either “less than 0.01” or “<0.01” for clarity. Corrected.

Round 2

Reviewer 2 Report

Comments and Suggestions for Authors

I appreciate the authors’ thorough revision. All previous concerns have been well addressed, and I have no additional comments.